# Extreme Values are Accurate and Robust in Deep Networks

## Abstract

Recent evidence shows that convolutional neural networks (CNNs) are biased towards textures so that CNNs are non-robust to adversarial perturbations over textures, while traditional robust visual features like SIFT (scale-invariant feature transforms) are designed to be robust across a substantial range of affine distortion, addition of noise, etc with the mimic of human perception nature. This paper aims to leverage good properties of SIFT to renovate CNN architectures towards better accuracy and robustness. We borrow the scale-space extreme value idea from SIFT, and propose EVPNet (extreme value preserving network) which contains three novel components to model the extreme values: (1) parametric differences of Gaussian (DoG) to extract extrema, (2) truncated ReLU to suppress non-stable extrema and (3) projected normalization layer (PNL) to mimic PCA-SIFT like feature normalization. Experiments demonstrate that EVPNets can achieve similar or better accuracy than conventional CNNs, while achieving much better robustness on a set of adversarial attacks (FGSM,PGD,etc) even without adversarial training.

## 1 Introduction

Convolutional neural networks (CNNs) evolve very fast ever since AlexNet (Krizhevsky & Hinton, 2012) makes a great breakthrough on ImageNet image classification challenge (Deng et al., 2009) in 2012. Various network architectures have been proposed to further boost classification performance since then, including VGGNet (Simonyan & Zisserman, 2015), GoogleNet (Szegedy et al., 2015), ResNet (He et al., 2016), DenseNet (Huang et al., 2017) and SENet (Hu et al., 2018), etc. Recently, people even introduce network architecture search to automatically learn better network architectures (Zoph & Le, 2017; Liu et al., 2018).

However, state-of-the-art CNNs are challenged by their robustness, especially vulnerability to adversarial attacks based on small, human-imperceptible modifications of the input (Szegedy et al., 2014; Goodfellow et al., 2015). Su et al. (2018) thoroughly study the robustness of 18 well-known ImageNet models using multiple metrics, and reveals that adversarial examples are widely existent. Many methods are proposed to improve network robustness, which can be roughly categorized into three perspectives: (1) modifying input or intermediate features by transformation (Guo et al., 2018), denoising (Liao et al., 2018; Jia et al., 2019), generative models (Samangouei et al., 2018; Song et al., 2018); (2) modifying training by changing loss functions (Wong & Kolter, 2018; Elsayed et al., 2018; Zhang et al., 2019), network distillation (Papernot et al., 2016), or adversarial training (Goodfellow et al., 2015; Tramer et al., 2018) (3) designing robust network architectures (Xie et al., 2019; Svoboda et al., 2019; Nayebi & Ganguli, 2017) and possible combinations of these basic categories. For more details of current status, please refer to a recent survey (Akhtar & Mian, 2018).

Although it is known that adversarial examples are widely existent (Su et al., 2018), some fundamental questions are still far from being well studied like what causes it, and how the factor impacts the performance, etc. One of the interesting findings in (Su et al., 2018) is that model architecture is a more critical factor to network robustness than model size (e.g. number of layers). Some recent works start to explore much deeper nature. For instance, both (Geirhos et al., 2019; Baker et al., 2018) show that CNNs are trained to be strongly biased towards textures so that CNNs do not distinguish objects contours from other local or even noise edges, thus perform poorly on shape dominating object instances. On the contrary, there are no statistical difference for human behaviors on both texture rich objects and global shape dominating objects in psychophysical trials. Ilyas et al. (2019) further analyze and show that deep convolutional features can be categorized into robust and

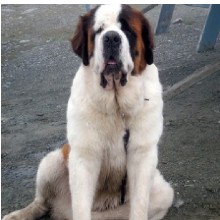 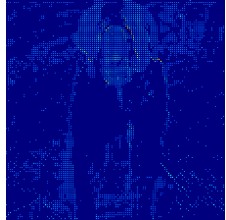 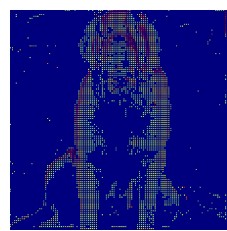

Figure 1: Left: Input image; Middle: response after ReLU of first conv-layer in ResNet-50; Right: response of first EVPConv-layer in EVPNet-50 (by replacing all $k \times k$ conv-layers in ResNet-50). Note multi-channel feature maps are merged into one channel with per-pixel max-out operations for easy illustration. Clearly, ResNet-50 has more noise responses, while EVPNet-50 gives more responses on object boundary.

non-robust features, while non-robust features may even account for good generalization. However, non-robust features are not expected to have good model interpretability. It is thus an interesting topic to disentangle robust and non-robust features with certain kinds of *human priors* in the network designing or training process.

In fact, human priors have been extensively used in handcraft designed robust visual features like SIFT (Lowe, 2004). SIFT detects scale-space (Lindeberg, 1994) extrema from input images, and selects stable extrema to build robust descriptors with refined location and orientation, which achieves great success for many matching and recognition based vision tasks before CNN being reborn in 2012 (Krizhevsky & Hinton, 2012). The scale-space extrema are efficiently implemented by using a difference-of-Gaussian (DoG) function to search over all scales and image locations, while the DoG operator is believed to biologically mimic the neural processing in the retina of the eye (Young, 1987). Unfortunately, there is (at least explicitly) no such scale-space extrema operations in all existing CNNs. Our motivation is to study the possibility of leveraging good properties of SIFT to renovate CNN networks architectures towards better accuracy and robustness.

In this paper, we borrow the scale-space extrema idea from SIFT, and propose extreme value preserving networks (EVPNet) to separate robust features from non-robust ones, with three novel architecture components to model the extreme values: (1) parametric DoG (pDoG) to extract extreme values in scale-space for deep networks, (2) truncated ReLU (tReLU) to suppress noise or non-stable extrema and (3) projected normalization layer (PNL) to mimic PCA-SIFT (Ke et al., 2004) like feature normalization. pDoG and tReLU are combined into one block named EVPConv, which could be used to replace all $k \times k$ ($k > 1$) conv-layers in existing CNNs. We conduct comprehensive experiments and ablation studies to verify the effectiveness of each component and the proposed EVPNet. Figure 1 illustrates a comparison of responses for standard convolution + ReLU and EVPConv in ResNet-50 trained on ImageNet, and shows that the proposed EVPConv produces less noises and more responses around object boundary than standard convolution + ReLU, which demonstrates the capability of EVPConv to separate robust features from non-robust ones. Our major contribution are:

- To the best of our knowledge, we are the first to explicitly separate robust features from non-robust ones in deep neural networks from an architecture design perspective.
- We propose three novel network architecture components to model extreme values in deep networks, including parametric DoG, truncated ReLU, and projected normalization layer, and verify their effectiveness through comprehensive ablation studies.
- We propose extreme value preserving networks (EVPNets) to combine those three novel components, which are demonstrated to be not only more accurate, but also more robust to a set of adversarial attacks (FGSM, PGD, etc) even for clean model without adversarial training.

## 2   RELATED WORK

**Robust visual features.** Most traditional robust visual feature algorithms like SIFT (Lowe, 2004) and SURF (Bay et al., 2006) are based on the scale-space theory (Lindeberg, 1994), while there is a close link between scale-space theory and biological vision (Lowe, 2004), since many scale-space operations show a high degree of similarity with receptive field profiles recorded from the mammalian retina and the first stages in the visual cortex. For instance, DoG computes the difference of two Gaussian blurred images and is believed to mimic the neural processing in the retina (Young, 1987). SIFT is one such kind of typical robust visual features, which consists of 4 major stages: (1) scale-space extrema detection with DoG operations; (2) Keypoints localization by their stability; (3) Orientation and scale assignment based on primary local gradient direction; (4) Histogram based

keypoint description. We borrow the scale-space extrema idea from SIFT, and propose three novel and robust architecture components to mimic key stages of SIFT.

**Robust Network Architectures.** Many research efforts have been devoted to network robustness especially on defending against adversarial attacks as summarized in Akhtar & Mian (2018). However, there are very limited works that tackle this problem from a network architecture design perspective. A major category of methods (Liao et al., 2018; Xie et al., 2019) focus on designing new layers to perform denoising operations on the input image or the intermediate feature maps. Most of them are shown effective on black-box attacks, while are still vulnerable to white-box attacks. Non-local denoising layer proposed in Xie et al. (2019) is shown to improve robustness to white-box attack to an extent with adversarial training (Madry et al., 2018). Peer sample information is introduced in Svoboda et al. (2019) with a graph convolution layer to improve network robustness. Biologically inspired protection (Nayebi & Ganguli, 2017) introduces highly non-linear saturated activation layer to replace ReLU layer, and demonstrates good robustness to adversarial attacks, while similar higher-order principal is also used in Krotov & Hopfield (2018). However, these methods still lack a systematic architecture design guidance, and many (Svoboda et al., 2019; Nayebi & Ganguli, 2017) are not robust to iterative attack methods like PGD under clean model setting. In this work, inspired by robust visual feature SIFT, we are able to design a series of innovative architecture components systematically for improving both model accuracy and robustness.

We should stress that extreme value theory is a different concept to scale-space extremes, which tries to model the extreme in data distribution, and is used to design an attack-independent metric to measure robustness of DNNs (Weng et al., 2018) by exploring input data distribution.

## 3 PRELIMINARY

**Difference-of-Gaussian.** Given an input image $I$ and Gaussian kernel $G(x, y, \sigma)$ as below

$$G(x, y, \sigma) = \frac{1}{2\pi\sigma^2} e^{-(x^2+y^2)/2\sigma^2}, \tag{1}$$

where $\sigma$ denotes the variance. Also, difference of Gaussian (DoG) is defined as

$$D(x, y, \sigma) = G(x, y, \sigma) \otimes I_1 - G(x, y, \sigma) \otimes I_0, \tag{2}$$

where $\otimes$ is the convolution operation, and $I_1 = G(x, y, \sigma) \otimes I_0$. Scale-space DoG repeatedly convolves input images with the same Gaussian kernels, and produces difference-of-Gaussian images by subtracting adjacent image scales. Scale-space extrema (maxima and minima) are detected in DoG images by comparing a pixel to its 26 neighbors in $3\times3$ grids at current and two adjacent scales (Lowe, 2004).

**Adversarial Attacks.** We use $h(\cdot)$ to denote the softmax output of classification networks, and $h^c(\cdot)$ to denote the prediction probability of class $c$. Then given a classifier $h(\mathbf{x}) = y$, the goal of adversarial attack is to find $\mathbf{x}^{\text{adv}}$ such that the output of classifier deviates from the true label $y$: $\max_i h^i(\mathbf{x}^{\text{adv}}) \neq y$ while closing to the original input: $||\mathbf{x} - \mathbf{x}^{\text{adv}}|| \leq \epsilon$. Here $|| \cdot ||$ refers to a norm operator, i.e. $L_2$ or $L_\infty$.

**Attack Method.** The most simple adversarial attack method is Fast Gradient Sign Method (FGSM) (Goodfellow et al., 2015), a single-step method which takes the sign of the gradient on the input as the direction of the perturbation. $L(\cdot, \cdot)$ denotes the loss function defined by cross entropy. Specifically, the formation is as follows:

$$\mathbf{x}^{\text{adv}} = \mathbf{x} + \epsilon \cdot \text{sign}(\nabla_{\mathbf{x}} L(h(\mathbf{x}), y)), \tag{3}$$

where $\mathbf{x}$ is the clean input, $y$ is the label. $\epsilon$ is the norm bound ($||\mathbf{x} - \mathbf{x}^{\text{adv}}|| \leq \epsilon$, i.e. $\epsilon$-ball) of the adversarial perturbation. Projected gradient descent (PGD) iteratively applies FGSM with a small step size $\alpha_i$ (Kurakin et al., 2017a; Madry et al., 2018) with formulation as below:

$$\mathbf{x}^{\text{adv}}_{i+1} = \text{Proj}(\mathbf{x} + \alpha \cdot \text{sign}(\nabla_{\mathbf{x}} L(h(\mathbf{x}_i^{adv}), y))), \tag{4}$$

where $i$ is the iteration number, $\alpha = \epsilon/T$ with $T$ being the number of iterations. 'Proj' is the function to project the image back to $\epsilon$-ball every step. Some advanced and complex attacks are further introduced in DeepFool (Moosavi-Dezfooli et al., 2016), CW (Carlini & Wagner, 2017), MI-FGSM (Dong et al., 2018).

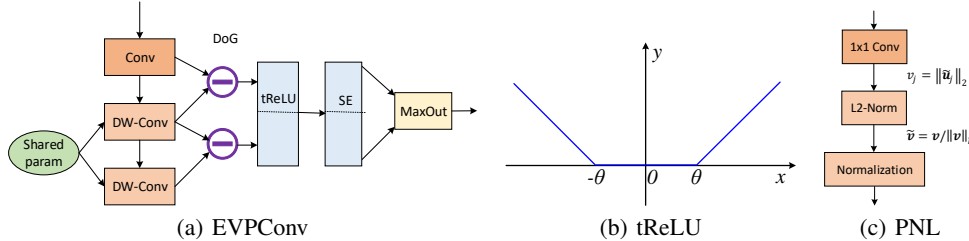

(a) EVPConv            (b) tReLU            (c) PNL

Figure 2: Novel components in EVPNet (a) Overall architecture of EVPConv; (b) plot of truncated ReLU (tReLU) function; (c) projected normalization layer (PNL). Here $\|\cdot\|_p$ means $\ell_p$ norm.

**Adversarial Training** aims to inject adversarial examples into training procedure so that the trained networks can learn to classify adversarial examples correctly. Specifically, adversarial training solves the following empirical risk minimization problem:

$$\arg\min_{h\in H} E_{(\mathbf{x},y)\ D}[\max_{\mathbf{x}^*\in A(\mathbf{x})} L(h(\mathbf{x}^*),y)], \tag{5}$$

where $A(\mathbf{x})$ denotes the area around $\mathbf{x}$ bounded by $L_\infty/L_2$ norm $\epsilon$, and $H$ is the hypothesis space. In this work, we employ both FGSM and PGD to generate adversarial examples for adversarial training.

## 4 METHOD

Inspired by traditional robust visual feature SIFT, this paper aims to improve model accuracy and robustness by introducing three novel network architecture components to mimic some key components in SIFT: parametric DoG (pDoG), truncated ReLU (tReLU), and projected normalization layer (PNL). Combining pDoG and tReLU constructs the so-called extreme value preserving convolution (EVPConv) block as shown in Figure 2(a), which can be used to replace all $k \times k$ ($k > 1$) conv-layers in existing CNNs. PNL is a new and robust layer plugged in to replace global average pooling (GAP) layer as shown in Figure 2(c). A network with all the three components is named as extreme value preserving network (EVPNet). In the following, we will describe these three components in details separately, and elaborate on how they are used to construct the EVPConv block and EVPNet.

### 4.1 THREE BASIC COMPONENTS TO MIMIC SIFT

**Parametric DoG (pDoG)** is a network component we design to mimic DoG operation. Recall DoG in Equation 2, it repeatedly convolves input images with the **same** Gaussian kernel in which kernel size $\sigma$ is designable, and then computes the differences for adjacent Gaussian blurred images. For CNNs, we mimic DoG with two considerations. *First*, we replace the Gaussian kernel with a learnable convolutional filter. Specifically, we treat each channel of feature map separately as one image, and convolve it with a learnable $k \times k$ kernel to mimic Gaussian convolution. Note that the learnable convolution kernel is not required to be symmetric since some recent evidence shows that non-symmetric DoG may perform even better (Einevoll & Plesser, 2012; Winnemöller, 2011). Applying the procedure to all the feature-map channels is equal to a depth-wise (DW) convolution (Howard et al., 2017). *Second*, we enforce successive depth-wise convolutions in the same block with shared weights since traditional DoG operation uses the same Gaussian kernel. As CNNs produce full scale information at different stages with a series of convolution and downsampling layers, each pDoG block just focuses on producing extrema for current scale, while not requiring to produce full octave extrema like SIFT. The shared DW convolution introduces minimum parameter overhead, and avoid "extrema drift" in the pDoG space so that it may help finding accurate local extrema. Formally, given input feature map $\mathbf{f}_0$, a minimum of two successive depth-wise convolution is applied as

$$\mathbf{f}_1 = DW(\mathbf{f}_0; \mathbf{w}); \qquad \mathbf{f}_2 = DW(\mathbf{f}_1; \mathbf{w}), \tag{6}$$

where $DW(;)$ is depth-wise convolution with $\mathbf{w}$ as the shared weights. pDoG is thus computed as

$$\mathbf{d}_0 = \mathbf{f}_1 - \mathbf{f}_0; \qquad \mathbf{d}_1 = \mathbf{f}_2 - \mathbf{f}_1 = DW(\mathbf{f}_1; \mathbf{w}) - DW(\mathbf{f}_0; \mathbf{w}). \tag{7}$$

It is worth noting that the successive minus operations make the minus sign not able to be absorbed into $\mathbf{w}$ for replacing minus into addition operation. To the best of our knowledge, this is the first time, minus component has been introduced into deep neural networks, which brings totally new element for architecture design/search.

Following SIFT, we compute local extrema (maxima and minimal) across the pDoG images using maxout operations (Goodfellow et al., 2013):

$$\mathbf{z}_0 = \max(\mathbf{d}_0, \mathbf{d}_1); \qquad \mathbf{z}_1 = -\min(\mathbf{d}_0, \mathbf{d}_1) = \max(-\mathbf{d}_0, -\mathbf{d}_1). \tag{8}$$

Note we do not compute local extrema in $3 \times 3$ spatial grids as in SIFT since we do not require keypoint localization in CNNs. Finally, to keep the module compatible to existing networks, we need ensure the output feature map to be of the same size (number of channels and resolution). Therefore, a maxout operation is taken over to merge two feature maps and obtain the final output of this block:

$$\mathbf{o} = \max(\mathbf{z}_0, \mathbf{z}_1). \tag{9}$$

**Truncated ReLU (tReLU).** The pDoG block keeps all the local extrema in the DoG space, while many local extrema are unstable because of small noise and even contrast changes. SIFT adopts a local structure fitting procedure to reject those unstable local extrema. To realize similar effect, we propose truncated ReLU (tReLU) to suppress non-robust local extrema. The basic idea is to truncate small extrema which correspond to noise and non-stable extrema in the pDoG space. This can be implemented by modifying the commonly used ReLU function as

$$y = \begin{cases} |x| & \text{if } |x| \geq \theta, \\ 0 & \text{if } |x| < \theta, \end{cases} \tag{10}$$

where $\theta$ is a learnable truncated parameter. Note that this function is discontinued at $x = \theta$ and $x = -\theta$. We make a small modification to obtain a continuous version for easy training as below

$$y = tReLU(x) = \begin{cases} |x| - \theta & \text{if } |x| \geq \theta, \\ 0 & \text{if } |x| < \theta. \end{cases} \tag{11}$$

Figure 2(b) plots the tReLU function. Different from the original ReLU, tReLU introduces a threshold parameter $\theta$ and keeps elements with higher magnitude. $\theta$ can be either a block-level parameter (each block has one global threshold) or a channel-level parameter (each channel holds a separate threshold). By default, we take $\theta$ as a block-level parameter.

tReLU is combined with pDoG not only to suppress non-robust extrema, but also to simplify the operations. When combining Equation 8 and Equation 9 together, there is nested maxout operation which satisfies commutative law, so that we could rewrite $\mathbf{z}_0$ and $\mathbf{z}_1$ as

$$\mathbf{z}_0 = \max(\mathbf{d}_0, -\mathbf{d}_0) = |\mathbf{d}_0|; \qquad \mathbf{z}_1 = \max(\mathbf{d}_1, -\mathbf{d}_1) = |\mathbf{d}_1|, \tag{12}$$

where $|\cdot|$ is element-wise absolute operation. With tReLU to suppress non-robust features, we have

$$\mathbf{z}_0 = tReLU(\mathbf{d}_0); \qquad \mathbf{z}_1 = tReLU(\mathbf{d}_1). \tag{13}$$

Hence, in practice, we use Equation 13 instead of Equation 8 to compute $\mathbf{z}_0$ and $\mathbf{z}_1$. Note that tReLU does improve robustness and accuracy for pDoG feature maps, while providing no benefits when replacing ReLU in standard CNNs according to our experiments (see Table 1).

**Projected Normalization Layer (PNL).** SIFT (Lowe, 2004) computes gradient orientation histogram followed by $L2$ normalization to obtain final feature representation. This process does not take gradient pixel relationship into account. PCA-SIFT (Ke et al., 2004) handles this issue by projecting each local gradient patch into a pre-computed eigen-space using PCA. We borrow the idea from PCA-SIFT to build projected normalization layer (PNL) to replace global average pooling (GAP) based feature generation in existing CNNs. Suppose the feature-map data matrix before GAP is $\mathcal{X} \in \mathbb{R}^{d \times c}$, where $d = w \times h$ corresponds to feature map resolution, and $c$ is the number of channels, we obtain column vectors $\{\mathbf{x}_i \in \mathbb{R}^c\}_{i=1}^d$ from $\mathcal{X}$ to represent the $i$-th pixel values from all channels. The PNL contains three steps:

(1) We add a $1 \times 1$ conv-layer, which can be viewed as a PCA with learnable projection matrix $\mathcal{W} \in \mathbb{R}^{c \times p}$. The output is $\mathbf{u}_i = \mathcal{W}^T \mathbf{x}_i$, where $\mathbf{u}_i \in \mathbb{R}^p$ further forms a data matrix $\mathcal{U} \in \mathbb{R}^{d \times p}$.

(2) We compute $L2$ norm for row vectors $\{\tilde{\mathbf{u}}_j \in \mathbb{R}^d\}_{j=1}^p$ of $\mathcal{U}$, to obtain a vector $\mathbf{v} = (v_1, \cdots, v_p)$ with $v_j = \|\tilde{\mathbf{u}}_j\|_2$.

(3) To eliminate contrast or scale impact, we normalize $\mathbf{v}$ to obtain $\tilde{\mathbf{v}} = \mathbf{v}/\|\mathbf{v}\|_p$, while $\|\cdot\|_p$ means the $\ell_p$ norm. the normalized vector $\tilde{\mathbf{v}}$ is fed into classification layer for prediction purpose.

It is interesting to note that PNL actually computes a second order pooling similar as (Gao et al., 2019; Yu & Salzmann, 2018). Suppose $\mathbf{w}_j \in \mathbb{R}^c$ is the $j$-th row of $\mathcal{W}$, $v_j$ in step-2 can be rewritten as

$$v_j = \sqrt{\sum_{i=1}^d (\mathbf{w}_j^T \mathbf{x}_i)^2} = \sqrt{\sum_{i=1}^d \mathbf{w}_j^T \mathbf{x}_i \mathbf{x}_i^T \mathbf{w}_j} = \sqrt{\mathbf{w}_j^T (\sum_{i=1}^d \mathbf{x}_i \mathbf{x}_i^T) \mathbf{w}_j} = \sqrt{\mathbf{w}_j^T A \mathbf{w}_j}, \tag{14}$$

where $A = \sum_{i=1}^d \mathbf{x}_i \mathbf{x}_i^T$ is an auto-correlation matrix. Figure 2(c) illustrates the PNL layer. Theoretically, GAP produces a hyper-cube, while PNL produces a hyper-ball. This is beneficial for robustness since hyper-ball is more smooth, and a smoothed surface is proven more robust (Cohen et al., 2019). Our experiments also verify this point (see Table 1).

| Network | +pDoG | +tReLU | +PNL | Clean | FGSM | PGD-10-1 | PGD-10-2 | DeepFool | CW |
|---|---|---|---|---|---|---|---|---|---|
| | | | | 91.68 | 24.23 | 3.03 | 0.00 | 20.19 | 0.57 |
| | ✓ | | | 92.55 | 31.14 | 3.37 | 0.00 | 29.33 | 0.60 |
| | | ✓ | | 91.38 | 25.37 | 2.79 | 0.00 | 23.41 | 0.23 |
| SE-ResNet-20 | | | ✓ | 90.92 | 40.66 | 13.16 | 1.66 | 31.86 | 1.02 |
| | ✓ | ✓ | | 91.98 | 37.46 | 4.64 | 0.01 | 36.54 | 0.98 |
| | ✓ | | ✓ | 92.62 | 60.12 | 28.03 | 6.37 | 39.28 | 4.56 |
| | ✓ | ✓ | ✓ | **92.85** | **66.21** | **41.06** | **12.19** | **40.28** | **6.78** |

Table 1: Ablation study results on CIFAR-10, attack with $\epsilon = 8$. Here 'PGD-$N$-$s$' denotes PGD attack with $N$ iterations of step size $s$ pixels.

## 4.2 EVPConv AND EVPNet

With these three novel components, we can derive a novel convolution block named EVPConv, and the corresponding networks EVPNet. In details, EVPConv starts from the pDoG component, and replaces Equation 8 with tReLU as in Equation 13. In SIFT, the contribution of each pixel is weighted by the gradient magnitude. This idea can be extended to calibrate contributions from each feature-map channel. Fortunately, Squeeze-and-Excitation (SE) module proposed in (Hu et al., 2018) provides the desired capability. We thus insert the SE block after tReLU, and compute the output of EVPConv as:

$$
\begin{aligned}
(\mathbf{s}_0, \mathbf{s}_1) &= SE(concat(\mathbf{z}_0, \mathbf{z}_1)), \\
\mathbf{o} &= \max(\mathbf{s}_0, \mathbf{s}_1),
\end{aligned}
\tag{15}
$$

where $concat(\cdot)$ means concatenating $\mathbf{z}_0$ and $\mathbf{z}_1$ together for a unified and unbiased calibration, $SE(\cdot)$ is the SE module, $\mathbf{s}_0$ and $\mathbf{s}_1$ are the calibration results corresponding to $\mathbf{z}_0$ and $\mathbf{z}_1$, and $\max$ denotes an element-wise maximum operation. Figure 2(a) illustrates the overall structure of EVPConv.

EVPConv can be plugged to replace any $k \times k$ ($k > 1$) conv-layers in existing CNNs, while the PNL layer can be plugged to replace the GAP layer for feature abstraction. The network consisting of both EVPConv block and the PNL layer is named as EVPNet. The EVPConv block introduces very few additional parameters: $\mathbf{w}$ for shared depth-wise convolution, $\theta$ for tReLU and parameters for SE module. Note that we allow each EVPConv block having its own $\mathbf{w}$ and $\theta$. EVPConv brings relatively fewer additional parameters, which is about 7∼20% (see Appendix A) (smaller models more relative increasing). It also increases theoretic computing cost 3∼10% for a bunch of parameter-free operations like DoG and maxout. However, the added computing cost is non-negligible in practice (2× slower according to our training experiments) due to more memory cost for additional copy of feature-maps. Near memory computing architecture (Singh et al., 2019) may provide efficient support for this new computing paradigm.

## 5 EXPERIMENTS

**Experimental Setup.** We evaluate the proposed network components and EVPNet on CIFAR10 and SVHN datasets. CIFAR-10 is a widely used image classification dataset containing $60,000$ images of size $32 \times 32$ with $50,000$ for training and $10,000$ for testing. SVHN (Netzer et al., 2011) is a digit recognition dataset containing 73,257 training images, 26,032 test images, all with size $32 \times 32$.

We introduce our novel components into the well-known and widely used ResNet, and compare to the basic model on both clean accuracy and adversarial robustness. As the EVPConv block contains a SE module, to make a fair comparison, we set SE-ResNet as our comparison target. In details, we replace the input conv-layer and the first $3 \times 3$ conv-layer in the residual block with EVPConv, and replace the GAP layer with the proposed PNL layer. Following (He et al., 2016; Huang et al., 2017), for CIFAR-10, all the networks are trained with SGD using momentum 0.9, 160 epochs in total. The initial learning rate is 0.1, divided by 10 at 80 and 120 epochs. For SVHN, we use the same network architecture as CIFAR-10. The models are trained for 80 epochs, with initial learning rate 0.1, divided by 10 at 40 and 60 epochs. For tReLU, the channel-level parameter $\theta$ is initialized by uniformly sampling from $[0, 1]$.

In this work, we consider adversarial perturbations constrained under $l_\infty$ norm. The allowed perturbation norm $\epsilon$ is 8 pixels (Madry et al., 2018). We evaluate non-targeted attack adversarial robustness in three settings: normal training, FGSM adversarial training (Goodfellow et al., 2015; Tramer et al., 2018) and PGD adversarial training (Madry et al., 2018). During adversarial training, we use the predicted label to generate adversarial examples to prevent label leaking effect (Kurakin et al., 2017b). To avoid gradient masking (Tramer et al., 2018), we use R-FGSM for FGSM adversarial training, which basically starts from a random point in the $\epsilon$ ball. Following Madry et al. (2018), during training, PGD attacks generate adversarial examples by 7 PGD iterations with 2-pixel step size starting from random points in the allowed $\epsilon$ ball. We report accuracy on both whitebox and blackbox

| Network | Training Method | Model | Clean | FGSM | PGD-10 | PGD-40 | DeepPool | CW | Blackbox |
|---|---|---|---|---|---|---|---|---|---|
| SE-ResNet-20 | Normal | Baseline | 91.68 | 24.23 | 0.00 | 0.00 | 20.19 | 0.57 | 78.32 |
| | | EVPNet | **92.85** | **66.21** | **12.19** | **6.84** | **40.28** | **6.78** | **79.31** |
| | FGSM | Baseline | 84.84 | 62.17 | 29.41 | 27.74 | 60.88 | 25.54 | 81.81 |
| | | EVPNet | **84.94** | **64.90** | **34.10** | **31.75** | **62.74** | **29.87** | **82.50** |
| | PGD | Baseline | 84.02 | 64.39 | 36.23 | 34.84 | 63.01 | 32.59 | 82.71 |
| | | EVPNet | **84.30** | **65.90** | **38.60** | **36.94** | **64.23** | **34.14** | **83.83** |
| SE-ResNet-56 | Normal | Baseline | **94.20** | 37.18 | 0.00 | 0.00 | 24.33 | 0.97 | 83.42 |
| | | EVPNet | 93.80 | **74.05** | **29.39** | **7.43** | **44.59** | **8.33** | **84.04** |
| | FGSM | Baseline | 87.50 | 67.88 | 35.84 | 33.87 | 62.34 | 34.54 | 86.17 |
| | | EVPNet | **89.61** | **69.93** | **41.79** | **35.63** | **63.45** | **35.42** | **87.57** |
| | PGD | Baseline | 86.86 | 69.45 | 39.90 | 39.90 | 64.22 | 38.90 | 87.33 |
| | | EVPNet | **90.13** | **71.33** | **40.81** | **40.80** | **66.11** | **39.10** | **89.00** |

Table 2: Comparison results on CIFAR-10 at different training settings and different networks, attack with $\epsilon = 8$.

| Network | Training Method | Model | Clean | FGSM | PGD-10 | PGD-40 | DeepFool | CW | Blackbox |
|---|---|---|---|---|---|---|---|---|---|
| SE-ResNet-20 | Normal | Baseline | 95.76 | 70.44 | 50.18 | 0.13 | 32.19 | 0.22 | 90.27 |
| | | EVPNet | **96.55** | **80.89** | **66.71** | **4.71** | **43.17** | **7.78** | **90.49** |
| | FGSM | Baseline | 95.79 | 66.53 | 5.83 | 1.94 | 56.33 | 4.94 | 90.56 |
| | | EVPNet | **96.44** | **90.80** | **43.17** | **20.71** | **58.22** | **18.23** | **90.78** |
| | PGD | Baseline | 92.47 | 74.72 | 6.25 | 1.53 | 57.56 | 6.36 | 91.03 |
| | | EVPNet | **96.06** | **80.45** | **31.30** | **8.69** | **60.02** | **12.43** | **92.40** |
| SE-ResNet-56 | Normal | Baseline | 96.48 | 75.14 | 1.19 | 0.05 | 34.92 | 0.43 | 90.53 |
| | | EVPNet | **96.68** | **84.60** | **41.94** | **14.30** | **46.46** | **9.94** | **90.94** |
| | FGSM | Baseline | 96.60 | 76.03 | 3.67 | 0.00 | 57.32 | 6.78 | 91.03 |
| | | EVPNet | **97.09** | **81.53** | **33.83** | **18.34** | **59.54** | **20.34** | **91.70** |
| | PGD | Baseline | 86.50 | 69.92 | 12.95 | 3.27 | 59.23 | 7.83 | 91.43 |
| | | EVPNet | **96.39** | **77.82** | **20.96** | **7.21** | **60.34** | **13.10** | **92.06** |

Table 3: Comparison results on SVHN at different training settings and different networks, attack with $\epsilon = 8$.

attack. We evaluate a set of well-known whitebox attacks, including FGSM, PGD, DeepFool, CW. We use 'PGD-$N$' to denote attack with $N$ PGD iterations of step size 2 pixels by default. Specifically, we compare results for PGD-10 and PGD-40. For blackbox attack, we choose VGG-16 as the source model which is found by Su et al. (2018) to exhibit high adversarial transferability, and choose FGSM as the method to generate adversarial examples from VGG-16 as it is shown to lead to better transferability Su et al. (2018).

**Ablation Study.** This part conducts a thorough ablation study to show the effectiveness of each novel architecture component and how they interact to provide strong adversarial robustness. We conduct experiments on CIFAR-10 with SE-ResNet-20, which contains one input conv-layer, three residual stage each with two bottleneck residual blocks, and GAP layer followed by a classification layer. We evaluate the accuracy and robustness for all the possible combinations of the proposed three components under the normal training setting. For PGD attack, we use two step sizes: 1 pixel as in Xie et al. (2019) and 2 pixels as in Madry et al. (2018). Table 1 lists full evaluation results of each tested model. Several observations can be made from the table:

(1) Solely adding pDoG or PNL layer leads to significant robustness improvement. pDoG even yields clean accuracy improvement, while PNL yields slightly clean accuracy drops.

(2) tReLU does not bring benefit for standard convolution, while yields notable improvement on both clean accuracy and adversarial accuracy, when combining with pDoG. That verifies our previous claim that tReLU is suitable to work for the DoG space.

(3) Combining all the three components together obtains the best adversarial robustness, while still achieve 1.2% clean accuracy improvement over the model without these three components.

Based on these observations, we incorporate all the three components into the CNNs to obtain the so-called EVPNet for the following experiments if not explicitly specified. As 2-pixels PGD attack is much stronger than 1-pixel PGD attack, we use it as default in the following studies.

**Benchmark Results.** We conduct extensive experiments on CIFAR-10 and SVHN to compare the proposed EVPNet with the source networks. The two sources networks are ResNet-20 and ResNet-56. For fair comparison, we use the SE extended ResNet as our baseline. Table 2 lists comprehensive comparison results on CIFAR-10. We list 7 different kinds of accuracies: clean model accuracy, whilebox attack accuracies by FGSM/PGD-10/PGD-40/DeepFool/CW, and blackbox attack accuracy with adversarial examples generated by FGSM on the VGG-16 model. We can see that under normal training case, EVPNet outperforms baseline by a large margin in terms of robustness with FGSM, PGD, DeepFool, and CW attacks. Even under the strongest PGD-40 white box attack, our EVPNet still has non-zero accuracy without any adversarial training. For those cases with adversarial training, our EVPNet consistently beats baseline networks with noticeable margin.

Table 3 further lists comparison results on SVHN. Similarly, our EVPNet consistently outperforms baseline models on all the three training settings. It is interesting to note that PGD adversarial

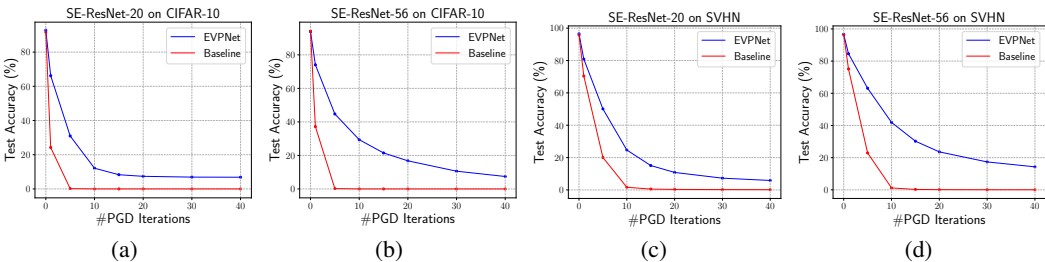

Figure 3: Test accuracy *vs* the number of PGD iterations for both networks on two datasets.

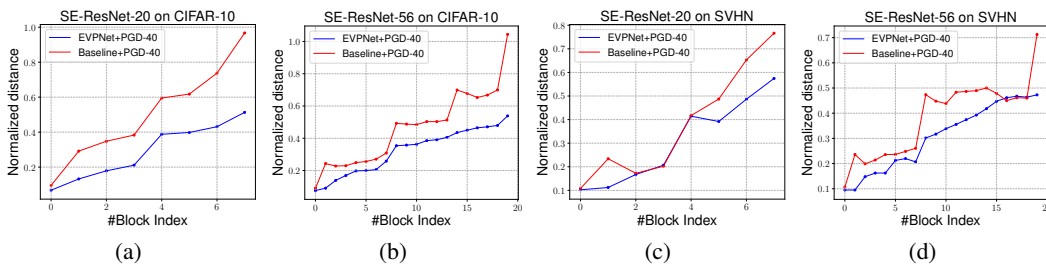

Figure 4: Normalized adversarial-benign distance for feature responses between adversarial and benign examples at different res-block output. Networks are trained without adversarial training.

training performs worse on PGD-10/PGD-40 attacks than FGSM adversarial training, and even much worse than normal training on this dataset. This may be due to the fact that SVHN is a shape/edge dominating digit recognition dataset, which may generate a lot of difficult adversarial samples with broken edges. And it also coincides with the finding by Baker et al. (2018). Our EVPNet shows better robustness on this dataset without adversarial training than CIFAR-10, which may suggest that EVPNet is more robust on shape/edge dominating object instances. All these evidences prove that the proposed EVPNet is a robust network architecture.

**Analysis.** We make some further analysis to compare EVPNet to baseline networks. *First*, we plot the test error at different PGD iterations for different evaluated networks under normal training case on both CIFAR-10 and SVHN datasets as shown in Figure 3. It can be seen that EVPNet consistently performs significantly better than the corresponding baseline networks under all PGD iterations. Some may concern that the accuracy of EVPNet on the strongest PGD-40 attack is not satisfied ($\sim 10\%$). We argue that from three aspects: (1) The adversarial result is remarkable as it is by the clean model without using any other tricks like adversarial training, adversarial loss, etc. (2) The proposed components also brings consistent clean accuracy improvement even on large-scale dataset (see Appendix A). (3) More importantly, the methodology we developed may shed some light on future studies in network robustness and network architecture design/search.

*Second*, we further investigate the error amplification effect as Liao et al. (2018). Specifically, we feed both benign examples and adversarial examples into the evaluated networks, and compute the normalized $L_2$ distance for each res-block outputs as $\gamma = \|\mathbf{x} - \mathbf{x}'\|_2 / \|\mathbf{x}\|_2$, where $\mathbf{x}$ is the response vector of benign example, and $\mathbf{x}'$ is the response vector for the adversarial example. We randomly sample 64 images from the test set to generate adversarial examples using PGD-40. The models evaluated are trained without adversarial training. Figure 4 illustrates the results. As we can see, EVPNet has much lower average normalized distance than the baseline models almost on all the blocks. It is interesting to see that the baseline models have a big jump for the normalized distance at the end of the networks on all the 4 sub-figures. This urges the adversarial learning researchers to make further investigation on the robustness especially for latter layers around GAP. Nevertheless, this analysis demonstrates that EVPNet significantly reduces the error amplification effect.

*Third*, we compare the differences on feature responses between regular convolution + ReLU and EVPConv. This comparison is made on large-scale and relative high resolution ($224 \times 224$) ImageNet dataset for better illustration. We train ResNet-50 and EVPNet-50 on ImageNet, and visualize their prediction responses for the first corresponding convolution block in Figure 1. It clearly shows that ResNet-50 has more noise responses, while EVPNet-50 gives more responses on object boundary. This demonstrates the capability of EVPConv to separate robust features from non-robust ones. Full benchmark results on ImageNet are also very promising, see Appendix A for more details.

## 6 CONCLUSION

This paper mimics good properties of robust visual feature SIFT to renovate CNN architectures with some novel architecture components, and proposes the extreme value preserving networks (EVPNet). Experiments demonstrate that EVPNets can achieve similar or better accuracy over conventional CNNs, while achieving much better robustness to a set of adversarial attacks (FGSM, PGD, etc) even for clean model without any other tricks like adversarial training.

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

## A EXPERIMENTS ON IMAGENET

We further make experiments on large-scale ImageNet classification dataset. We propose to replace the widely used ResNet architecture with the proposes components, and obtain the derived network named EVP-ResNet. We did experiments on three ResNet architectures: ResNet26A/26B and ResNet50. ResNet26A/26B are two variants of ResNet18, both having the same number of residual blocks (ResBlock) as ResNet18, but replacing standard ResBlock with bottleneck ResBlock. ResNet26A has the same filter width in each stage as ResNet18 (i.e, [64, 64, 128, 256, 512]). Due to the bottleneck structure, ResNet26A has just about 1/8 parameters of ResNet18. ResNet26B has the same filter width in each stage as ResNet50 (i.e., [64, 256, 512, 1024, 2048]), which yields just slightly higher parameter size than ResNet18, but much lower than ResNet50. Note there are two small modifications. *First*, we move the SE module in SE-ResNet after the last conv-layer in the bottleneck ResBlock to after the 2nd conv-layer, which is consistent with our EVPConv block for fair comparison. *Second*, for ResNet, we implement the stride = 2 convolution in downsampling ResBlock as ResNet-B structure in He et al. (2019).

Besides ResNet, we also try to extend our framework on MobileNet-v1 (Howard et al., 2017), which is a plane network structure without residual connections. We take standard MobileNet-V1 (1.0× in channel width) as baseline, and provide experiments comparison to the SENet extension and the EVPNet extension. For our EVPNet extension, we add EVPConv module after each $3 \times 3$ depth-wise convolution, and replace the GAP layer with our PNL layer. For the SENet extension, we add SE module after each $3 \times 3$ depth-wise convolution for consistency with the EVPNet extension.

For fair comparison, We trained all the models (ResNet and extensions, MobileNet and extensions) by ourselves on the same GPU server with 8 NVidia 2080TI GPUs with the same training configurations: (1) the same data augmentation (baseline one used by He et al. (2019));(2) Nesterov Accelerated Gradient (NAG) descent as the optimizer; (3) training 90 epochs with batchsize 256; (4) the learning rate is initialized to 0.1 and divided by 10 at the 30th, 60th, and 80th epochs. We evaluated model performance on the validation set with just single center-crop test, without using any data augmentation in this procedure. Besides clean model accuracy, we also evaluate the adversarial attacks by FGSM and PGD-10 with maximum perturbation for each pixel $\epsilon = 8$ under $l_\infty$ norm.

Table 4 lists the full comparison results as well as model information like parameter size and computing cost. It shows that EVP-ResNet gives consistent accuracy improvement over ResNet and SE-ResNet in all the three tested network architectures. EVP-ResNet26A improves ResNet26A and SE-ResNet26A by 6.09% and 3.61% respectively in absolute accuracy. EVP-ResNet50 improves ResNet50 and SE-ResNet50 by 1.97% and 0.83% respectively in absolute accuracy.

| Network | Model | Clean | FGSM | PGD-10 | #Params(M) | FLOPs(G) |
|---|---|---|---|---|---|---|
| ResNet26A | Baseline | 57.63 | 21.45 | 0.10 | 1.33 | 0.261 |
| | SE-ResNet | 60.11 | 26.98 | 0.45 | 1.35 | 0.262 |
| | EVP-ResNet | **63.72** | **30.23** | **6.78** | 1.65 | 0.290 |
| ResNet26B | Baseline | 72.39 | 28.31 | 0.10 | 15.24 | 2.34 |
| | SE-ResNet | 74.14 | 32.19 | 1.02 | 15.58 | 2.36 |
| | EVP-ResNet | **75.38** | **38.12** | **9.45** | 17.01 | 2.41 |
| ResNet50 | Baseline | 75.04 | 29.13 | 0.23 | 24.36 | 4.10 |
| | SE-ResNet | 76.18 | 33.45 | 1.20 | 24.97 | 4.11 |
| | EVP-ResNet | **77.01** | **40.12** | **8.30** | 26.75 | 4.18 |
| MobileNet-v1 | Baseline | 70.07 | 18.45 | 0.09 | 4.04 | 0.557 |
| | SE-MobileNet | 71.51 | 30.92 | 0.87 | 4.70 | 0.560 |
| | EVP-MobileNet | **72.63** | **37.54** | **6.99** | 5.48 | 0.623 |

Table 4: Top-1 accuracy by Resnet26A/B, Resnet50, and MobileNet on ImageNet under normal training. The last two columns list the model parameter size and model computing FLOPs.

In terms of parameter size, EVP-ResNet brings 7∼20% additional parameters to the ResNet counterparts (smaller models more relative increasing). In terms of computing FLOPs, EVP-ResNet increases 3∼10% for a bunch of parameter-free operations like DoG and maxout. However, the added computing cost is non-negligible in practice (2× slower in our training experiments) due to more memory cost for additional copy of feature-maps.

In the adversarial attack evaluation, EVP-ResNet still consistently outperforms the counterparts of ResNet, SE-ResNet. For the strongest PGD-10 attacks, all ResNet and SE-ResNet models drop

top-1 accuracy to near zero, while the EVP-ResNet variants keep 6~10% top-1 accuracy. The gap in FGSM attacks is even larger. This improvement is remarkable considering that it is by clean model without adversarial training.

For the MobileNet case, we also observe notable accuracy and robustness improvement. Please refer to Table 4 for more details. In summary, our solid results and attempts may inspire future new ways for robust network architecture design or even automatic search.

# B  WIDE-RESNET EXPERIMENTS ON CIFAR-10

In the main paper, we demonstrate the great robustness of the proposed components on ResNet with bottleneck residual blocks. Here, we extend the proposed components to other state-of-the-art network architectures, and choose Wide-ResNet (Zagoruyko & komodakis, 2017) as an example since it is mostly studied on other adversarial training works (Madry et al., 2018; Cisse et al., 2017; Athalye et al., 2018). Wide-ResNet (WRN) has two successive wide-channel $3 \times 3$ conv-layers in residual block instead of the three conv-layer bottleneck structure based residual block. We use WRN-22-8 as the baseline network with depth 22 and widening factor 8. It is obvious that WRN-22-8 has much better clean accuracy than ResNet-20 and ResNet-56 used in the main paper. For EVPNet, We replace input conv-layer and the first $3 \times 3$ conv-layer in each wide residual block with EVPConv, and replace the GAP layer with our PNL layer. Table 5 shows the comparison on normal training case. We can see that EVPNet achieves similar clean accuracy, while performing significantly better on adversarial attacks with FGSM/PGD-10/PGD-40. Note that under PGD-10 and PGD-40 attacks, the baseline model drops accuracy to near 0, while the EVPNet remains a much higher accuracy, considering that no adversarial training is utilized in this study. This demonstrates the strong robustness of the proposed EVPNet.

| Network | Training Method | Model | Clean | FGSM | PGD-10 | PGD-40 | Blackbox |
|---------|-----------------|-------|-------|------|--------|--------|----------|
| SE-WRN-22-8 | Normal | Baseline | 95.84 | 49.34 | 0.01 | 0.00 | 83.45 |
|  |  | EVPNet | **95.90** | **79.48** | **43.07** | **22.14** | **84.51** |

Table 5: Results by SE-WRN-22-8 on CIFAR-10, attack with $\epsilon = 8$.

