# OpenReview forum: "Extreme Values are Accurate and Robust in Deep Networks"
_ICLR.cc/2020/Conference — Reject_

### Official Review · AnonReviewer3 · 2019-10-21
**Official Blind Review #3**

**Rating:** 3

**Review:**

In this paper, a new network architecture called EVPNet was proposed to improve robustness of CNNs to adversarial perturbations. To this end, EVPNet employs three methods to leverage scale invariant properties of SIFT features in CNNs.

The proposed network and the methods are interesting, and provide promising results in the experiments. However, there are several issues with the paper:

- The authors claim that Gaussian kernels are replaced by convolution kernels to mimic DoGs. However, it is not clear (1) how this replacement, or employment of convolution kernels can mimic DoGs, or (2) more precisely, how the corresponding learned convolution kernels approximate Gaussian kernels. In order to verify and justify this claim, please provide detailed theoretical and experimental analyses.

- It is also claimed that “a 1 × 1 conv-layer, can be viewed as a PCA with learnable projection matrix”. However, this statements is not clear. How do you assure that a 1x1 conv layer  employs a PCA operation or the corresponding projection?

- What does \| \|_p denote? Does it denote \ell_p norm?

- What does x denote in d = w x h? Previously, it was used to denote matrix size.

- Why do you compute \ell_2 norm for row vectors instead of column vectors? How do the results change when they are calculated for column vectors?

- According to the notation, s_0 and s_1 are vectors. Then, what does max denote in (14)? That is, how do you compute max(s_0, s_1), more precisely?

- In the statement “PNL produces a hyper-ball in the manifold space”, what do you mean by the “manifold space”? What are the structures (e.g. geometry, metrics etc.) and members of this space?

- Please conceptually and theoretically compare the proposed method with state-of-the-art methods following similar motivation, such as the following:

Weng et al., Evaluating the Robustness of Neural Networks: An Extreme Value Theory Approach, ICLR 2018.


**Experience Assessment:**

I have published in this field for several years.

**Review Assessment: Checking Correctness Of Derivations And Theory:**

I carefully checked the derivations and theory.

**Review Assessment: Checking Correctness Of Experiments:**

I assessed the sensibility of the experiments.

**Review Assessment: Thoroughness In Paper Reading:**

I read the paper at least twice and used my best judgement in assessing the paper.

---

> ### Author Response · Authors · 2019-11-13
> **Response to Reviewer #3**
>
> Thank the reviewer for the helpful feedback and suggestions.
>
> 1.    ## Details on How pDOG replace DoG ##
> In DoG, the operation is shown in Eq-2,
>                 $D(x,y,\sigma) = G(x, y, \sigma) \otimes I_1 – G(x,y, \sigma) \otimes I_0$.
> Where $\sigma$ is pre-designed Gaussian kernel size, $\otimes$ means convolution the kernel with input image, and $I_1 = G(x,y, \sigma) \otimes I_0$.
>
> In the pDoG, we just mimic the filtering process in Eq-7 with
>         	$d_1 = DW(f_1, w) – DW(f_0, w)$,
> where DW is depth-wise convolution, $w$ is the parameters of DW, and $f_1 = DW(f_0, w)$.
> It is obviously that the pDoG replace the Gaussian blur in DoG with a learnt convolutional filter, and each channel is processed separately, so that the operation could be modeled with a depth-wise convolution.
> We want to stress that pDoG is inspired from DoG and mimic the processing, but it does not approximate Gaussian distribution at all.  The successive filtering with same learnt kernel will produce a scale-space, and we want to find the stable pixels/keypoints in the scale-space.  We hope that pDoG are able to learn better transformations than traditional DoG.
>
> 2. ## 1x1 convolution and PCA ##
> It is commonly known that 1x1 convolution is in actually MLP, as stated in the earliest work for 1x1 convolution (network-in-network, NIN) [1], while MLP is known that could be used to learn PCA projection matrix from data [2]. Here by “a PCA with learnable projection matrix”, we are stressing that 1x1 convolution is similar to PCA as it can be viewed as a dimensionality projection process with learnable projection matrix. Eq-14 (revised version) further explores how PCA data covariance matrix “A” is formulated in PNL.
>
>  3. ## Notation ##
>     - “\| \|_p”: Here we mean \ell_p norm.
>     - “d= w x h”: Here we mean ‘d’ is the product of ‘w’ and ‘h’, where ‘w’ and ‘h’ refers to the size/resolution of a feature map before the GAP layer. This means we reshape one 2D feature-map channel into a vector, and thus reshape the whole feature map from a 3D tensor into a 2D matrix with size $d \times c$, where c is the number of channels.
>     - “max(s_0, s_1)”: Here we mean it is maxout operation, which does per-element maximum.
> We have revised the description of those notations in the updated version.
>
> 4. ## why $\ell_2$ norm for row vectors, why not on column vectors ##
> Computing $\ell_2$ norm on row vectors of feature map, will output a vector of dimension $c$ , where $c$ is the number of feature map channels. This is consistent with global average pooling. If normalized on column vector, it will output a vector of dimension $d = w\times h$, i.e., the resolution of feature map. This will make the network not able to produce fixed length feature vector for different resolution input images, and requires additional overhead to handle on this issue.
>
> 5. ## manifold space of PNL ##
> The normalization after 1x1 convolution in PNL will push different data samples with feature representation $v$ distributed on a hyper-ball, while the GAP (global average pooling) is actually L_1 normalization, which will push data samples with GAP feature presentation distributed on a hyper-cubic. Here we just mean that these two have different geometric structure as manifolds.  We will revise the description accordingly.
>
> 6. ## Comparison with [3] ##
> Thanks for pointing out that paper.  First, we would like to emphasize that scale-space extreme values and extreme value theory are two different concepts/theory and developed indecently. Extreme value theory tries to model the extreme, rare events of the “data distributions” in certain kinds of functional space. While scale-space extreme values are biological inspired for the object boundary, keypoints, etc for “one input image”. Second, more specifically, [2] leverages more on data distribution extremes to propose an attack-independent metric CLEVER to measure robustness of neural network classifier. While, we focus on extract extremes in feature maps. Hence, there are no explicitly connections between these two concepts, and between our work and [3].
>
> [1] Network In Network. Lin et al. ArXiv 1312.4400.
> [2] Neural networks for pattern recognition, C. Bishop, Oxford Press, 1995.
> [3] Evaluating the Robustness of Neural Networks: An Extreme Value Theory Approach. Weng et al. ICLR 2018.

---

### Official Review · AnonReviewer1 · 2019-10-24
**Official Blind Review #1**

**Rating:** 3

**Review:**

This paper presents a SIFT-feature inspired modification to the standard convolutional neural network (CNN). Specifically the authors propose three innovations: (1) a differences of Gaussians (DoG) convolutional filter; (2) a symmetric ReLU activation function (referred to as a truncated ReLU; and (3) a projected normalization layer. The paper makes the claim that the proposed CNN variant (referred to as the EVPNet) demonstrates superior performance as well as improved robustness to adversarial attacks.

Clarity: Overall, the paper is not particularly well written. There are multiple missing articles and other grammatical errors that make it a bit arduous to read, though I do not believe they have obstructed my ability to understand the contributions. The section describing the projected normalization layer (second half of page 5) is a bit confusing. Figure 2(c) is not helpful in shedding light on the details, though I think a more detailed figure could be quite helpful. Beyond these issues, the paper is relatively clear in the presentation of the material.

Novelty: Over the last few years there have been many, many proposals for how to vary the basic CNN architecture to improve performance. Some of these have lead to genuine performance gains and have become part of the standard CNN specification. ReLUs, ResNets and Batch Normalization are particularly prominent examples of contributions that have been shown to lead to improvements in performance. Yet the vast majority of these sorts of proposals ultimately make little or no impact on the field. In light of this, I would rate the novelty of the basic goal of this paper as relatively low, though the specific proposal is novel to me and seems reasonable.

Impact: The impact potential for this paper lies with the performance offered by the proposed innovations. With respect to overall performance improvement the proposed method has not been shown to perform quite at a state-of-the-art level, as given by these resource:
SVHN: https://paperswithcode.com/sota/image-classification-on-svhn
CIFAR-10: https://paperswithcode.com/sota/image-classification-on-cifar-10

The authors compare the performance of their proposed EVPNet against a fair baseline - a squeeze-and-excite ResNet model. These sorts of controlled experiments are useful, but the actual reported performance for both models are somewhat off of the state-of-the-art and it's not clear that the relatively small benefit the authors show over their baselines are maintained for higher performing architectural configurations. Can this architecture be competitive with the state-of-the-art? The current paper in it's current

Most of the results relate to the claim that the proposed model is robust to adversarial examples. Unfortunately, this is not a particular area of expertise for me, so it's difficult for me to provide a confident assessment of the contribution here, though I will say two things: (i) the method seems to provide a significant increase in adversarial robustness across the
baseline architectures investigated. (ii) the authors demonstrate that the benefit provided by the proposed architecture seems to persist even when training for adversarial defence is introduced.

I would have liked to see more datasets explored in Experiments section. I  especially would have liked to see results on ImageNet.

My current rating is weak reject based on the weakness of the writing and the lack of strong empirical evidence in support of the effectiveness of the proposed contributions.

**Experience Assessment:**

I have published in this field for several years.

**Review Assessment: Checking Correctness Of Derivations And Theory:**

I carefully checked the derivations and theory.

**Review Assessment: Checking Correctness Of Experiments:**

I assessed the sensibility of the experiments.

**Review Assessment: Thoroughness In Paper Reading:**

I read the paper thoroughly.

---

> ### Author Response · Authors · 2019-11-13
> **Response to Reviewer #`1**
>
> Thank you for the detailed feedback and suggestions and we are happy to address your concerns.
>
> 1. ## Clarity ##
> Thanks for pointing out about the writing issue. We will fix the grammatical errors in our revision and improve the presentation of Figure 2.
>
> 2. ## Novelty ##
> We would like to point out that our work is significant different to most existing CNN architecture design works in three aspects.
> First, to the best of our knowledge, we are the first to bring biologic inspired scale-space theory for traditional robust visual features like SIFT into CNN architecture design. Especially, the successive minus operations make the minus sign (-) not able to be absorbed into $\mathbf{w}$ in Eq-7 for replacing minus into addition operation. To the best of our knowledge, this is the first time, minus component has been introduced into deep neural networks, which brings totally new element for architecture design/search.
> Second, we are the first to consider full network architecture design from adversarial robustness perspective, rather than from clean accuracy or efficiency perspective. It is also quite different to majority of works on adversarial robustness, such as adversarial training [1, 2], input/feature map de-noising [3, 4]. Our designed network shows not only better clean accuracy but also much better adversarial robustness on a bunch of dataset (CIFAR-10, SVHN, ImageNet).
> Furthermore, the feature map by EVPConv illustrated in Figure-1 is also more meaningful and explainable.
> We believe our work is different to those works the reviewer referred to as “ultimately make little or no impact on the field.” We also believe that biologic-inspired scale-space theory is quite fundamental and promising direction for CNN architecture design.
>
> 3. ## Impact by state-of-the-art networks ##
> We use current ResNet architecture, as it was widely used in existing studies [2,4,5,6]. Per the suggestion, we also conduct experiments to test the performance on more deeper or wider state-of-the-art networks like Wide-ResNet [7]. The results are shown in Appendix-B. We can clearly see that even clean accuracy is much higher, EVPNet still brings significant robustness improvement over the baseline, while still keep similar clean accuracy.
>
> 4. ## Results on more datasets like ImageNet ##
> In fact, in the original submission, we already include results with ResNet on ImageNet in Appendix-A. We further include more results for the Mobilenet-v1 (without residual connection) structure. All demonstrates effectiveness of the proposed method.
>
> [1] Explaining and Harnessing Adversarial Examples. Ian J. Goodfellow, Jonathon Shlens, Christian Szegedy. ICLR 2015.
> [2] Towards Deep Learning Models Resistant to Adversarial Attacks. Aleksander Madry, Aleksandar Makelov, Ludwig Schmidt, Dimitris Tsipras, Adrian Vladu. ICLR 2018.
> [3] Defense against adversarial attacks using high-level representation guided denoiser. Fangzhou Liao, Ming Liang, Yinpeng Dong, Tianyu Pang, Xiaolin Hu, Jun Zhu. CVPR 2018.
> [4] Feature Denoising for Improving Adversarial Robustness. Cihang Xie, Yuxin Wu, Laurens van der Maaten, Alan Yuille, Kaiming He. CVPR 2019.
> [5] PixelDefend: Leveraging Generative Models to Understand and Defend against Adversarial Examples. Yang Song, Taesup Kim, Sebastian Nowozin, Stefano Ermon, Nate Kushman. ICLR 2018.
> [6] Theoretically Principled Trade-off between Robustness and Accuracy. Hongyang Zhang, Yaodong Yu, Jiantao Jiao, Eric P. Xing, Laurent El Ghaoui, Michael I. Jordan. ICML 2019.
> [7] Wide Residual Networks. Sergey Zagoruyko, Nikos Komodakis. BMVC, 2016.

---

### Official Review · AnonReviewer2 · 2019-11-01
**Official Blind Review #2**

**Rating:** 8

**Review:**

This paper proposes a network model named EVPNet, inspired by the idea scale-space extreme value from SIFT, to improve network robustness to adversarial pertubations over textures. To achieve better robustness, EVPNet separates outliers (non-robust) from robust examples by extenting DoG to parametric DoG, utilising truncated ReLU, and then applying a projected normalisation layer to mimic PCA-SIFT like feature normalisation, which are the three novelties that the authors claim in this paper. In the experiments, FGSM and PGD are used to provide adversarial attacks, and experiments conducted on CIFAR-10 and SVHN reveal that EVPNet enhances network robustness.
Overall, this paper contributes to network robustness from an architecture perspective; in the contrast, most prior works focus more on robust feature extraction and loss function design. The ablation study of EVPNet demonstrates the effectiveness of its each novel component. A further investigation presents that EVPNet reduces the error ampplification effects. The example that the authors show in this paper demonstrates the improvement of EVPNet to image textures.

The reviewer has some main concerns regarding the claimed novelty:
1. pDoG computes the difference between outputs of two depth-convolution layers, but there is no evidence that the distribution of feature maps is gaussian or gaussian-like. There is no clarification for this point.

2. Truncated ReLU is a modified ReLU. Does the learnable truncated parameter \theta limit its applicability to different datasets?

3. The Projected Normalisation Layer (PNL) seems a reasonable implementation, but essentially it is not very different from batch normalisation. The authors state only its difference to global average pooling but not to batch normalisation which should be a better comparison.

For the experiments, the following should be addressed:

4. Experiments were conducted only for the SE-ResNet architecture via replacing its CNN kernel by the proposed EVPNet. Although SE-ResNet shows good performance on some common data, but the squeeze-excitation block might bring in non-robustness. Hence, the reviewer thinks that it is risky to claim: the replacement of EVPNet in CNN layers is robust to adverserial attacks based only on this implementation. Try EVPNet for a more basic network architecture (VGG) would be suggested.

5. The \epsilon, which represents the adversarial attack tolerance, is always 8. There is no explaination in this paper, why not other values.

Minor comments:
6. The authors did not clarify why the first novel component is called "parametric" DoG. There is a more "non-parametric" block.


**Experience Assessment:**

I have read many papers in this area.

**Review Assessment: Checking Correctness Of Derivations And Theory:**

I carefully checked the derivations and theory.

**Review Assessment: Checking Correctness Of Experiments:**

I carefully checked the experiments.

**Review Assessment: Thoroughness In Paper Reading:**

I read the paper at least twice and used my best judgement in assessing the paper.

---

> ### Author Response · Authors · 2019-11-13
> **Response to Reviewer #2**
>
> Thank the reviewer for the helpful feedback and suggestions.
>
> 1. ## pDoG: no evidence that feature maps are Gaussian ##
> DoG is not targeting for modeling the distribution of whole feature maps, but for local pixels (like 3x3) in the scale-space, and ensure to find stable pixels/keypoints across a series Gaussian blur of input image. So it does not require feature map to be Gaussian distribution. Our pDoG further extends DoG to make local filter without Gaussian assumption (see Eq-2 vs Eq-7 in the revised version). For each feature map channel, we view it as one input image, and blur that image successively with the same learnt filter, so that the series of blurred images consist of a scale-space, and we could find stable pixels/keypoints across all scales. Hence, we only mimic DoG for the filtering process rather than the filter itself.
>
> 2. ## truncated ReLU applied to other datasets ##
> We have tested our framework on CIFAR-10, SVHN, and large-scale ImageNet dataset. The results demonstrate that the tReLU module is generally effective and applicable in our EVPNet framework.
>
> 3. ## PNL vs BN ##
> PNL is a 1x1 projection layer followed by L2 normalization, while BN is “depth-wise 1x1 layer” (aka the scaling layer) followed by a covariance shifting for each pixel. The major difference is that PNL will project each channel to a scalar after L2 normalization and the whole tensor to a vector, while BN will not change the shape of input tensor.
>
> 4. ## EVPNet for other network ##
> For time limitation, we are not able to train VGGNet on ImageNet. However, we extend the novel blocks into the MobileNet (v1 without residual connection, very similar to VGG). Experimental results are added to Appendix-A. It shows that EVP-MobileNet still demonstrate much better accuracy and robustness comparing to original MobileNet and SE-MobileNet. We also provide experiments on Wide-ResNet in Appendix-B, which does not have bottleneck structures as ResNet in our experiments.
>
> 5. ## why epsilon always being 8 ##
> On CIFAR-10, \epsilon = 8 is a common choice in many adversarial attack studies [1,2,3,4,5].
> We also clarify the definition of epsilon-ball just after Eq-3.
>
> 6. ## parametric or non-parametric ##
> Here we talk parametric which means the filter parameters are learnt from data in pDoG, while in traditional DoG, the filter parameter is designed and fixed to be Gaussian function. Please also see Eq-2 and Eq-7 for the difference. The notion is somewhat different from the concept parametric model and non-parametric model in statistics.
>
> [1] Towards Deep Learning Models Resistant to Adversarial Attacks. Aleksander Madry, Aleksandar Makelov, Ludwig Schmidt, Dimitris Tsipras, Adrian Vladu. ICLR 2018.
> [2] Defense against adversarial attacks using high-level representation guided denoiser. Fangzhou Liao, Ming Liang, Yinpeng Dong, Tianyu Pang, Xiaolin Hu, Jun Zhu. CVPR 2018.
> [3] Feature Denoising for Improving Adversarial Robustness. Cihang Xie, Yuxin Wu, Laurens van der Maaten, Alan Yuille, Kaiming He. CVPR 2019.
> [4] PixelDefend: Leveraging Generative Models to Understand and Defend against Adversarial Examples. Yang Song, Taesup Kim, Sebastian Nowozin, Stefano Ermon, Nate Kushman. ICLR 2018.
> [5] Theoretically Principled Trade-off between Robustness and Accuracy. Hongyang Zhang, Yaodong Yu, Jiantao Jiao, Eric P. Xing, Laurent El Ghaoui, Michael I. Jordan. ICML 2019.

---

### Author Response · Authors · 2019-11-13
**Revision uploaded**

Thanks all the reviewers for helpful comments and suggestions.
The updated version includes following revisions to accommodate reviewers' concern.
(1) Revise $\epsilon$-ball definition in page-3 to make it consistent to the symbol used in experimental parts.
(2) Revise Eq-7 with expansion so that it looks more consistent with DoG definition. Add a note below to describe an important property.
(3) Add definition for the $\ell_p$ norm in Figure-2 and corresponding body part.
(4) Add more results on ImageNet at Appendix-A, especially results by the plane structure MobileNet-v1 and our extensions.
(5) Add results for CIFAR-10 by Wide-ResNet. The accuracy is now on-par with state-of-the-art results on CIFAR-10.
(6) Add a comparison to "Extreme value theory" and the paper suggested by reviewer-#3.
(7) Some grammar error and typos fix.

---

### Decision · Program_Chairs · 2019-12-19

**Decision:**

Reject

**Comment:**

This manuscript proposed biologically-inspired modifications to convolutional neural networks including differences of Gaussians convolutional filter, a truncated ReLU, and a modified projected normalization layer. The authors' results indicate that the modifications improve performance as well as improved robustness to adversarial attacks.

The reviewers and AC agree that the problem studied is timely and interesting, and closely related to a variety of recent work on robust model architectures. However, this manuscript also received quite divergent reviews, resulting from differences in opinion about the novelty and importance of the results. In reviews and discussion, the reviewers noted issues with clarity of the presentation and sufficient justification of the approach and results. In the opinion of the AC,  the manuscript in its current state is borderline and could be improved with more convincing empirical justification.